# Prenylated Flavonoids with Potential Antimicrobial Activity: Synthesis, Biological Activity, and In Silico Study

**DOI:** 10.3390/ijms22115472

**Published:** 2021-05-22

**Authors:** Mauricio Osorio, Marcela Carvajal, Alejandra Vergara, Estefania Butassi, Susana Zacchino, Carolina Mascayano, Margarita Montoya, Sophia Mejías, Marcelo Cortez-San Martín, Yesseny Vásquez-Martínez

**Affiliations:** 1Laboratorio de Productos Naturales, Departamento de Química, Universidad Técnica Federico Santa María, Valparaíso 2390123, Chile; 2Centro de Biotecnología CB-DAL, Universidad Técnica Federico Santa María, Valparaíso 2390136, Chile; marcela.carvajal@usm.cl (M.C.); alejandrajj@gmail.com (A.V.); 3Farmacognosia, Facultad de Ciencias Bioquímicas y Farmacéuticas, Universidad Nacional de Rosario, Suipacha 531, Rosario 2000, Argentina; fefabutassi@hotmail.com (E.B.); szaabigil@gmail.com (S.Z.); 4Departamento de Ciencias del Ambiente, Facultad de Química y Biología, Universidad de Santiago de Chile, Santiago 9170022, Chile; carolina.mascayano@usach.cl; 5Laboratorio Bioquímica Celular, Departamento de Biología, Facultad de Química y Biología, Universidad de Santiago de Chile, Santiago 9170022, Chile; margarita.montoya@usach.cl (M.M.); sofia.mejias@usach.cl (S.M.); 6Laboratorio de Virología Molecular y Control de Patógenos, Departamento de Biología, Facultad de Química y Biología, Universidad de Santiago de Chile, Santiago 9170022, Chile; marcelo.cortez@usach.cl; 7Programa Centro de Investigaciones Biomédicas Aplicadas, Escuela de Medicina, Facultad de Ciencias Médicas, Universidad de Santiago de Chile, Santiago 9170022, Chile

**Keywords:** prenylated flavonoids, synthesis, antibacterial, MRSA, synergism, antifungal

## Abstract

Prenylated flavonoids are an important class of naturally occurring flavonoids with important biological activity, but their low abundance in nature limits their application in medicines. Here, we showed the hemisynthesis and the determination of various biological activities of seven prenylated flavonoids, named **7–13**, with an emphasis on antimicrobial ones. Compounds **9**, **11**, and **12** showed inhibitory activity against human pathogenic fungi. Compounds **11**, **12** (flavanones) and **13** (isoflavone) were the most active against clinical isolated *Staphylococcus aureus* MRSA, showing that structural requirements as prenylation at position C-6 or C-8 and OH at positions C-5, 7, and 4′ are key to the antibacterial activity. The combination of **11** or **12** with commercial antibiotics synergistically enhanced the antibacterial activity of vancomycin, ciprofloxacin, and methicillin in a factor of 10 to 100 times against drug-resistant bacteria. Compound **11** combined with ciprofloxacin was able to decrease the levels of ROS generated by ciprofloxacin. According to docking results of *S* enantiomer of **11** with ATP-binding cassette transporter showed the most favorable binding energy; however, more studies are needed to support this result.

## 1. Introduction

Prenylated flavonoids are a subclass of natural flavonoids characterized by the presence of a prenylated side chain (i.e., prenyl, geranyl and lavandulyl) attached to the flavonoid skeleton. Since prenylated flavonoids are generally more bioactive than their non-prenylated precursors, they are of interest as lead compounds for producing new drugs [1,2,3]. However, since prenylated flavonoids often exist at trace levels in their natural sources that make further biological studies difficult, it may be useful to synthesize prenylated flavonoids in appreciable quantities and at low costs to allow future biological studies [4]. Compounds of this type exhibit significant effects in the insect–plant interaction [5] as well as antifungal [6], antimicrobial [7], antiviral [8,9], anti-inflammatory [10], and anticancer [11] activities and anti-lipid properties, both in vitro and in vivo [12]. The C-prenylation of flavonoids seems to be crucial to the biological activities of these compounds, which may lead to enhanced cell membrane targeting and thus increased intracellular activity [13]. From a chemistry point of view, the C-prenylation is complicated, inefficient, and time-consuming, so other methods must be explored; i.e., biocatalysis may be the solution to these problems [14,15]. The prenylation of the flavonoid core increases lipophilicity and membrane permeability, which is one of the proposed reasons for the enhanced biological activities of prenylated flavonoids [16]. Efficient chemistry methods to obtain 8-prenylnaringenin with reasonable yields (20–33%) have been reported via Claisen and Claisen–Cope rearrangement [17] in many steps. A good approximation to obtain these kinds of compounds may be the direct prenylation of commercially available flavonoids with 3-methyl-2-buten-1-ol using ZnCl_2_ as Lewis acid by coupling via an aromatic electrophilic substitution reaction (ArES), whose conditions have been used to obtain prenylated phenols in our laboratory [18].

To determine the structural requirements of the prenylated flavonoids to exert any remarkable biological activity, it is necessary to obtain numerous and varied series of these compounds and to test them against several types of biological tests. These results will provide enough data to obtain a reliable structure–activity relationship [17].

Multi-drug-resistant organisms are one of the major causes for the alarming level of infectious diseases worldwide [19]. The discovery of new drugs with potent antimicrobial activity, particularly against the resistant strains, is therefore highly desirable. Since many of the currently available drugs have undesirable side effects, are ineffective against new or reemerging fungi, or develop a rapid resistance [20], there is an urgent need for the next generation of new antimicrobial agents that overcome these disadvantages. In this sense, prenylated flavonoids have been demonstrated to have potent antimicrobial activity [21]. *Escherichia coli*, *Klebsiella pneumoniae*, and *Staphylococcus aureus* are the three human pathogens of greatest concern, associated with hospital- and community-acquired infections, according to the latest global estimates of antibiotic resistance worldwide [22].

Synergism between natural compounds and traditional antibiotics is a promising strategy that could solve the problem of antibiotic resistance that some bacterial strains have been able to develop [23].

As endogenous free radicals and human 5-lipoxygenase (5-hLOX) have been shown to have major roles in the pathogenesis of various health problems including inflammatory disorders [24,25], the compounds obtained here were tested as 5-hLOX inhibitors and DPPH (2,2-diphenyl-1-picrylhydrazyl) radical scavengers. Since inflammation is related to bacterial-mediated infections, a dual anti-inflammatory, antibacterial agent with an improved safety profile is a goal for improved therapeutic benefits and better patient compliance [26].

In this work, we have synthesized seven prenylated flavonoids and studied their antimicrobial, anti-inflammatory, and antioxidant activities to evaluate the relevance of adding prenyl groups on the structure of non-prenylated flavonoids and to propose the use of naturally occurring organic compounds as possible therapeutic agents.

## 2. Results and Discussion

### 2.1. Synthesis

The synthesis of prenylated phenols was carried out in one step via the EAS mechanism as shown in Scheme 1, using a prenylation methodology applied to hydroxylated acetophenones and benzaldehydes by us [18], with some modifications. The one-step reaction of different flavonoids (**1**–**5**) with 3-methyl-2-buten-1-ol (**6**) in the presence of ZnCl_2_ in ethyl acetate produced prenylated flavonoids (**7**–**13**) in moderate yields (23–36%) and side products, mainly. Only 5′-prenyl fisetin (**10**) is new. When the flavonoids chrysin (**1**) and naringenin (**4**) were alkylated, we isolated two C-alkylated compounds at positions 8 and 6 of the A-ring. Complex mixtures were found with quercetin (**2**) and fisetin (**3**), probably because their electron-donating substituents are more susceptible to oxidation [27]. Electron-rich aromatic rings can induce polyalkylations and decompositions, reducing the yield.

#### Structure Determination

The place of C-alkylation was determined by 2D ^1^H- ^13^C HMBC NMR spectra. The main *J_3_* and *J_2_* interactions between the methylene proton C*H*_2_ of the prenyl chain and aromatic peaks are schematically shown in Figure 1. Only the interaction between the methylene protons with the quaternary 8a carbon in the structure occurs when the C-prenylation is produced in C-8. Correlations *J_3_* and *J_2_* between the methylene protons C*H*_2_ and quaternary 8a or 4a carbon are impossible when the prenylation is bounded at Position 6. For the rest of the molecules, the same reasoning was used to determine the place of C-alkylation.

The presence of the carbonyl and hydroxyl groups was confirmed by observing the typical signals in IR spectra. Stretching absorption values between 3550 and 3200 cm^−1^ as a broad signal indicated the presence of phenol groups and stretching absorption values between 1750 and 1680 cm^−1^ as one of the strongest singlet IR absorptions confirmed the presence of carbonyl groups.

The hydrogen bond to carbons was determined by *J_1_* interactions in the 2D ^1^H-^13^C HSQC NMR spectra. 2D NMR spectra (HSQC and HMBC) for all prenylated compounds were included in Appendix A.

### 2.2. In Vitro Antibacterial Effects on Human Pathogen Bacteria

From the results of bacterial growth inhibitory activity on sensitive and resistant *S. aureus* (Table 1), we can observe that not all prenylated flavonoids showed a good antibacterial effect, being prenylated flavanones **7** to **10**, weak to moderate inhibitors. In Table 2, compounds **11** and **12**, corresponding to prenylated naringenin, together with compound **13** (prenylated genistein), showed strong activity both on the sensitive strain and on the resistant strains, with MIC values ranging from 5 to 50 μg/mL. Compound **11** is the one that shows the best antibacterial action, especially on the resistant strain MRSA 97-7 with an MIC lower than that of the commercial agents tested.

The results show that the saturation of the double bond at the C2–C3 position appears to be important for the antibacterial activity of flavonoids (compounds **11** and **12**). This coincides with those reported in the literature, where prenylated flavanones show better activity than flavones [28,29]. The position of the hydroxyl substituents, as well as that of the prenyl group, appears to be important in the inhibitory activity of bacterial growth, observing that derivatives with OH at positions C-5, 7, and 4′ show better inhibitory activity, as well as the presence of prenyl groups in the C6 and C8 positions of Ring A, as previously reported [30,31,32]. Araya–Coultier et al. (2018) showed that the position of the prenyl group affects the antimicrobial activity of flavonoids, when comparing the effect of lupiwighteone (5,7,4′-trihydroxy-8-prenylisoflavone) and wighteone (5,7,4′-trihydroxy-6-prenylisoflavone), on the Gram-positive strain *L. monocytogenes*. Lupiwighteone, chain prenylated at C-8, showed no inhibition, whereas wighteone, chain prenylated at position C-6, had high antibacterial activity [33].

When comparing compound **4** with compounds **11** and **12**, where the base structure is maintained and only one prenyl substituent has been added in different positions, we can observe that the introduction of a prenyl group has an important effect on the inhibitory activity, reducing by up to at least one order of magnitude the MIC in MRSA 97-7 (Table 2). It has been reported that prenylated flavonoids are more hydrophobic than common flavonoids, facilitating this characteristic the ability to penetrate the cell membrane, thus improving their action at the active site [29].

Mun et al. (2014) proposed that prenylated flavonoids act through interaction with the bacterial membrane, binding directly with peptidoglycan, observing morphological changes when MRSA strains were treated with the prenylated flavonoid sophora-flavanone B [34].

Table 3 and Table 4 show that a synergistic effect is produced between compounds **11**, **12**, and **13** when acting in conjunction with commercial antibiotics, showing that the molecules **11** and **12** reduce the MIC of ciprofloxacin over MRSA 97-7 by 10 times (FICI values 0.6 and 0.16, respectively) and **11**, **12**, and **13** reduce 100 times its MIC over MRSA 622-4 with FICI values from 0.1 to 0.2 (results shown in Table 4). They also acted synergistically with Vancomycin (FICI values from 0.15 to 0.43), decreasing its MIC 10 times against MRSA 97-7. Compound **11** also acted synergistically with Methicillin with an FICI value of 0.01, decreasing its MIC 100 times against MRSA 97-7.

This potentiating effect of the antimicrobial agent action exerted by prenylated molecules is not observed when combining them with each other, as observed in Table 5, where, when combining compounds **11** and **12**, an FICI value between 1.1 and 2 was achieved, given an account of an indifferent effect. This difference in the behavior of prenylated molecules, when compared to the action they exert when combined with antimicrobial agents, may be due to the fact that prenylated flavonoids act by a different mechanism than the commercial antibiotics evaluated, e.g., through the inhibition of pumps of expulsion such as ABC transporters, thus being able to enhance the action of the commercial antibiotic, as proposed by Wagner et al. (2009), who pointed out that the synergistic effect of phytodrugs in conjunction with antimicrobial agents can contribute to effects on the permeability of the cell wall of bacteria, the inhibition of the efflux pump, and the inhibition of bacterial enzymes, and that all these mechanisms would help the antimicrobial agent to reach its site of action [35].

To corroborate the synergistic action of compound **11** with vancomycin, a time kill assay experiment was carried out, the result of which is shown in Figure 2. We can see that compound **11** alone manages to slightly lower the bacterial growth curve, decreasing by less than 2 log the CFU/mL, showing a moderate bacteriostatic action. Vancomycin achieved a greater effect, lowering CFU/mL by more than 3 log10, but this effect is enhanced by acting in conjunction with compound **11**, lowering CFU/mL by more than 5 log10, thus verifying the synergistic effect that compound **11** would exert on ciprofloxacin on MRSA 97-7. It has been described in the literature that prenylated flavonoids isolated from *Morus alba* L. show a broad spectrum of synergistic action towards *Staphylococcus aureus*, acting in conjunction with antibacterial agents such as aminoglycosides, β-lactams, glycopeptides, and fluoroquinolones [28].

The results of the bacterial ROS level are shown in Figure 3. We can see that there is a significant decrease in ROS levels inside the bacteria as compounds **11**, **12**, and **13** are present, showing a protective effect by reducing ROS, with an effect as important as that shown by the agent Trolox, a known ROS species trapper. It has been described in the literature that polyhydroxylated flavonoids can act as antioxidant agents, trapping ROS [36], and this antioxidant property can be related to the ROS-reducing effect observed in this assay.

It has been previously reported that ciprofloxacin generates oxidative stress in *S. aureus*, generating an increase in ROS levels within the bacteria [37]. This result was corroborated in this study, where the levels of ROS generated in the bacteria in the presence of ciprofloxacin were statistically higher than those of the control. As a way to explain the synergistic action observed in the previous experiments, the effect on intrabacterial ROS levels was evaluated by combining ciprofloxacin with compound **11**. As we can see in Figure 4, this combination markedly decreases the levels of ROS when compared to those generated by ciprofloxacin alone. From these results, it can be deduced that the prenylated compounds may be acting as trappers for the ROS generated by ciprofloxacin.

#### Docking of **11** (S and R Enantiomers) and **8** with ATP-Binding Cassette (Multi-Drug ABC Transporter Sav1866 from *S. aureus* in Complex with AMP-PNP)

We used a fast and predictive tool as the molecular docking, which allowed us to obtain by the genetic algorithm the most suitable location of ligand on macromolecule without using molecular dynamic simulations. Thus, we carried out the analysis according to the most favorable binding energy, by grouping the results in a cluster. To analyze the good result of **11** and their correlation with the antibacterial activity, we used the in silico tool with a known and essential pharmacological target linked to this property. For the above, the ABC transporter was selected, which is crucial for balancing nutrients in bacteria and humans. This transporter is also responsible for several processes, such as drug-resistant bacteria [38]. Based on these reasons, we used docking tools to observe which interactions were relevant, especially with the most promising molecule **11**, although it is worth taking into account that the biological tests were carried out with a racemic mixture. In this study, we decided to study both enantiomers in order to be able to discriminate them and determine whether the binding was stereoselective. The results showed a priority for the *S* enantiomer (−5.06 Kcal/mol ± 0.012) versus the *R* enantiomer (−3.37 Kcal/mol ± 0.016) (a difference of approximately 2 Kcal/mol between them) both binding in the similar place of the transporter. These analyses of results effectively indicated a different binding mode: The *S* enantiomer showed two hydrogen bonds with Glu320 and Asp321 of the ABC transporter; on the other hand, the *R* enantiomer showed two hydrogen bonds but with Glu353 and Gly378 (Figure 5), and in both molecules, the hydrophobic interaction is important to the binding. We carried out a similar analysis by docking studies with **8** which has a bad antibacterial activity. The binding energy (-2.39 Kcal/mol) obtained was worse than the **11** enantiomers and the molecule only showed one hydrogen bond with Asp220 for stabilization in the binding area. Finally, we observed that the molecule is located in the same area as the isomers as we observed in Figure 5.

### 2.3. In Vitro Antifungal Effects on Human Pathogenic fungi

In order to have a look into the potential usefulness of these compounds as candidates for the development of new antifungal drugs, we investigated the antifungal properties of compounds **1**–**5** and **7**–**13** against a panel of clinically important fungal species including two yeasts (*Candida albicans* and *Cryptococcus neoformans*), three *Aspergillus* spp. (*A. niger, A. fumigatus,* and *A. flavus*), and three dermatophytes (*Trichophyton rubrum, T. mentagrophytes,* and *Microsporum gypseum*) (voucher numbers are included in Table 6). The minimum inhibitory concentration (MIC) and minimum fungicidal concentration (MFC) of all compounds were determined with the microbroth dilution methods M27-A3 and M38-A2 of the Clinical and Laboratory Standards Institute (CLSI) [39,40]. The evaluation of the minimum fungicidal concentration (MFC) of the prenylated flavonoids showed whether the compounds kill fungi in addition to inhibiting them. This characteristic is highly appreciated in an antifungal drug since the recurrence of fungistatic drugs is mostly avoided because long treatments are no longer necessary.

Compounds with MICs of >250 µg/mL were considered inactive; those between 250 and 125 µg/mL were considered lowly active; those in the range 62.5–31.25 µg/mL were considered moderately active; and MICs of <31.25 µg/mL were considered highly active.

The results showed (Table 6) that the non-prenylated structures (the flavone **1**, the flavonols **2** and **3**, the flavanone **4**, and the isoflavone **5**) were all inactive.

When these flavonoids were prenylated, the following results were obtained: (i) The prenylation of the flavone **1** or the isoflavone **5** did not improve the antifungal activity, since **7**, **8**, and **13** were inactive (MIC ≥ 250 µg/mL). (ii) The prenylation of the flavonols **2** and **3** led to **9** (which showed activity against the yeasts *C. albicans* and *C. neoformans* and dermatophytes at 62.5 µg/mL) and **10**, which was inactive. Both compounds possessed the prenyl substituent at position 5′ but differed in that **9** possesses a hydroxyl group in position 5. This structural difference could play a role in the antifungal activity of **9**. (iii) The prenylation of the flavanone **4** led to both **11** (MIC = 125 µg/mL against *C. neoformans* and 62.5 µg/mL against dermatophytes) and the most active compound **12**, which displayed MIC = 125 µg/mL against *C. albicans*, MIC = 62.5 µg/mL against *C. neoformans,* MIC = 15.6 µg/mL against the dermatophytes *T. rubrum* and *T. mentagrophytes*, and 31.2 µg/mL against *M. gypseum*). Probably the prenyl substituent at position 8 of the trihydroxyflavanone naringenin confers high activity to the whole flavanone molecule.

Considering that *C. neoformans* is the main cause of cryptococcal meningoencephalitis among HIV patients, which many times has not responded to any antifungal and has led to death [41], we decided to test compounds **9**, **11**, and **12** (which showed some activity against the ATCC strain) against a panel of five clinical *C. neoformans* strains. The results showed (Table 7) that the behavior of the three compounds against clinical isolates is similar to that against the standard ATCC strain.

In turn, since *C. albicans* is the fourth leading cause of nosocomial bloodstream infection (BSI) in intensive care units, causing fatal invasive candidiasis in a high percentage of patients [42], we decided to test **9** and **12** against five clinical isolates of *C. albicans*, in order to determine whether these two compounds have the potential to be further developed. The results showed (Table 8) that both compounds behave against *C. albicans* clinical isolates similarly to how they behave against the standard ATCC strain.

### 2.4. In Vitro Antibacterial Effects on Plant Pathogen bacteria

The results of the antibacterial activity against phytopathogenic bacteria showed that all the prenylated compounds were able to inhibit the in vitro growth of the three bacteria tested. Interestingly, most of the compounds significantly inhibited the growth of the three bacterial species in high percentages, obtaining IC_50_ values of less than 3.9 μM for several of them. Despite the above considerations, the **3**, **4**, and **5** molecules showed the highest IC_50_ values in the series (Table 9). Statistical analysis revealed that the values obtained from inhibition are significant (*p* < 0.05) compared to the negative control.

Antibacterial activities are shown as IC_50_ and MIC values in μM concentrations; IC_50_ values were calculated from the first kinetic assay. MIC: minimum inhibitory concentration.

IC_50_ values for each compound indicated that the bacteria tested had different levels of sensitivity to the compounds tested. The most sensitive bacteria were *P. syringae*, and the most resistant were *A. tumefaciens*. This behavior could be due to the different mechanisms of action of the compounds on each tested Gram-negative pathogen.

After the first kinetic test, the bactericidal capacity of each compound was evaluated in a second growth kinetics test, using an inoculum of the first kinetic plate [43]. This new culture was incubated for 7 h, and the absorbance was measured. It was observed that the bacterial culture was able to recover its growth rate (with respect to the culture without compound) by 50–85%. This means that the compounds were bacteriostatic but not bactericide at the concentrations tested. Compound **13** showed selectivity against *P. syringae* and *P. carotovorum* and to a lesser extent against *A. tumefaciens (*>250, MIC).

### 2.5. In Vitro 5-hLOX Enzyme Inhibition Assay

The results showed that, of all the evaluated molecules (Table 10), only two (compounds **9** and **10**) had a relevant inhibitory effect. The structural characteristic that differentiates these compounds from the rest of the structures is the catechol group (1,2-dihydroxybenzene) in Ring B. Many known inhibitors of 5-LOX possess the catechol group that is a relevant characteristic for their inhibitory properties [44].

### 2.6. DPPH Radical Scavenging Activity

The results obtained from antioxidant activity assays are shown in Table 11. All compounds were compared with Trolox. All compounds possessing the catechol moiety were active. It is known that the flavonoids quercetin (**2**) and fisetin (**3**) have better antioxidant activity than Trolox [45]; however, their prenylated derivatives (**9** and **10**) show an even greater degree of this activity. This behavior follows the same trend found in some of our previous work [18].

Antioxidant activity is shown as IC_50_ values in µM concentrations; NA = no activity. All compounds were analyzed in triplicate, and the results expressed as average ± standard deviation.

### 2.7. Cytotoxic Activity of **11** and **12**

Anticancer activity for naringenin in diverse tumor cell lines has been previously reported [46,47,48]. However, in vitro activities were commonly observed using high flavonoid concentrations often exceeding 100 µM, which have limited physiological significance and must be interpreted with caution. Given this anticancer activity and that prenylated flavonoids have enhanced biological activities, we measured cytotoxic activity prenylated naringenin at positions 6 and 8 (**11** and **12**) against MDA-MB-231 (the human breast adenocarcinoma cell line), B16-F10 (mouse melanoma cells), and MEF (primary mouse embryonic fibroblasts). Our results in Table 12 show that the cytotoxic effect of both prenylated naringenins were low for both cancer cell lines and did not show any selectivity compared to the effect on non-cancerous cells (MEFs). When the cytotoxic effect of **11** and **12** against cell lines were compared with commonly used chemotherapeutic drugs, it could be verified that both prenylated flavonoids showed from 100- to 250-fold lower activity than Taxol and from 30- to 50-fold lower activity than etoposide. It is possible that the anticancer activity described for naringenin in vivo could be related to antiangiogenic and immunostimulating effects [49,50].

On the other hand, compounds **11** and **12** exhibited the most effective antibacterial activity, which had a synergistic effect with commercial antibiotics. A significant synergic activity was observed using approximately 7 µM for **11** and 15 µM for **12** (Table 4), which did not generate a cytotoxic effect in MEF cells. This clearly suggests that **11** and **12** could be useful for in vivo treatment against pathogenic resistant bacteria, although further studies are needed to confirm this possibility.

## 3. Materials and Methods

### 3.1. Chemistry

#### 3.1.1. General Data

All chemical reagents obtained were purchased from Merck (Darmstadt, Germany), Sigma–Aldrich (St. Louis, MO, USA), or Alfa Aesar (Kandel, Germany), were of the highest commercially available purity, and were used without previous purification. Melting points (mp: °C) were measured on a melting point apparatus Stuart-Scientific SMP3 and are uncorrected. IR spectra were recorded as a KBr disk in a Thermo Scientific Nicolet 6700 FT-IR spectrometer (Massachusetts, USA), and frequencies are reported in cm^−1^. High-resolution mass spectra were recorded on an Exactive^TM^ Plus Orbitrap spectrometer (Thermo Scientific, Bremen, Germany) by infusion, applying a voltage of 5 or 8 kV in the positive ionization mode and 4 or 7 kV in the negative ionization mode. The spectra were recorded using full-scan mode, covering a mass range from m/z 214 to 470. The resolution was set to 140,000, and the maximum loading time for the ion cyclotron resonance (ICR) cell was set to 200 ms. ^1^H-, ^13^C-(DEPT 135), 2D HSQC, and 2D HMBC spectra were recorded in DMSO-*d*_6_ solutions and referenced to the residual peaks of DMSO at δ 2.50 ppm for ^1^H and δ 39.5 ppm for ^13^C, respectively, on a Bruker Avance 400 Digital NMR spectrometer, operating at 400.1 MHz for ^1^H and 100.6 MHz for ^13^C. Chemical shifts are reported in δ ppm and coupling constants (*J*) are given in Hz. Silica gel (Merck 200–400 mesh) was used for C.C. and silica gel plates HF-254 for TLC. TLC spots were detected by both under a UV lamp and heating after drenching in 10% H_2_SO_4_ in H_2_O. Antioxidant determinations were performed in a Thermo Scientific Multiskan GO 96-well plate photometer.

#### 3.1.2. General Experimental Procedure for the Prenylation of Flavonoids

A solution of flavonoid (1 mol equiv) and dry ZnCl_2_ (4 mol equiv) was placed in a round bottom flask and dissolved in ethyl acetate (100 mL). Under vigorous stirring, a solution of 3-methyl-2-buten-1-ol (**6**) (4 mol equiv) in ethyl acetate (10 mL) was added dropwise for 1 h at 40 °C. The reaction mixture was then allowed to warm up to reflux temperature and the stirring continued. After 4 h, water at pH 1 (100 mL) was added to the reaction mixture to decompose the ZnCl_2_. The organic layer was then separated, and a new extraction with ethyl acetate was made. The organic solutions obtained after extractions were mixed and dried over anhydrous sodium sulphate and filtered, and the solvent was evaporated under reduced pressure. The mixture was then subjected to silica gel flash column chromatography (ethyl acetate/hexane or dichloromethane/methanol or acetone/hexane mixtures were used as mobile phases) to obtain pure products.

All structures were confirmed by IR and NMR spectra as discussed below.

#### 3.1.3. Physical Data of Prenylated Flavonoids

2-phenyl-8-(3-methyl-2-buten-1-yl)-5,7-dihydroxy-4*H*-chromen-4-one (**7**) and 2-phenyl-6-(3-methyl-2-buten-1-yl)-5,7-dihydroxy-4*H*-chromen-4-one (**8**) were obtained from chrysin (**1**) (3.9 mmol), 3-methyl-2-buten-1-ol (**6**) (15.7 mmol), and ZnCl_2_ (15.7 mmol), as described above. The crude mixture was purified using ethyl acetate-hexane in a gradient system (0 → 70% of ethyl acetate) as the mobile phase to afford **7** as a yellow powder (329 mg, 26%); mp: 222 °C; HRMS *m*/*z*, observed: 321.1131; C_20_H_17_O_4_ [M–H]^−^ requires: 321.1127. IR (KBr): ν_max_ cm^−1^: 3431, 2960, 2927, 2858, 1731, 1647, 1615, 1579, 1507, 1384, 1363, 1275. ^1^H-NMR (DMSO-*d*_6_) δ ppm: 12.75 (s, 1H, ArO*H*-5); 10.6–11.0 (br. s, 1H, ArO*H*-7); 8.03 (d, 2H, *J* = 6.8 Hz, 2′,6′-Ar*H*); 7.59–7.57 (m, 3H, 3′,4′,5′-Ar*H*); 6.95 (s, 1H, Ar*H*-3); 6.30 (s, 1H, Ar*H*-6); 5.18 (br. t., 1H, C*H* = C(CH_3_)_2_); 3.44 (d, 2H, *J* = 6.7 Hz, C*H*_2_CH = C(CH_3_)_2_); 1.74 (s, 3H, -CH_2_CH = CC*H*_3_CH_3_); 1.61 (s, 3H, -CH_2_CH = CCH_3_C*H*_3_). ^13^C-NMR (DMSO-*d*_6_) δ ppm: 17.8 (-CH = CCH_3_CH_3_); 21.3 (-CH_2_CH = CCH_3_CH_3_); 25.4 (CH_2_CH = CCH_3_CH_3_); 98.5 (ArC-6); 103.9 (ArC-4a); 105.0 (ArC-3); 106.3 (ArC-8); 122.4 (CH = C(CH_3_)_2_); 126.3 (ArC-2′,6′); 129.2 (ArC-3′,5′); 131.1 (CH = C(CH_3_)_2_); 132.0 (ArC-4′); 154.6 (ArC-8a); 159.1 (ArC-5); 161.9 (ArC-7); 163.1 (Ar*C*-2); 182.2 (4-C = O).

Compound **8** was obtained as a yellow powder (291 mg, 23%) mp: 221–222 °C (lit. [51] 213–214 °C). HRMS *m*/*z*, observed: 321.1130; C_20_H_17_O_4_ [M–H]^−^ requires: 321.1127. IR (KBr): ν_max_ cm^−1^: 3400, 2969, 2918, 2651, 1639, 1580, 1553, 1483, 1449; ^1^H-NMR (DMSO-*d*_6_) δ ppm: 13.08 (s, 1H, ArO*H*-5); 8.05 (d, 2H, *J* = 7.1 Hz, 2′,6′-Ar*H*); 7.60–7.56 (m, 3H, 3′,4′,5′-Ar*H*); 6.95 (s, 1H, Ar*H*-3); 6.57 (s, 1H, Ar*H*-8); 5.17 (br. t, 1H, C*H* = C(CH_3_)_2_); 3.22 (d, 2H, *J* = 6.8 Hz, -C*H*_2_CH = C(CH_3_)_2_); 1.72 (s, 3H, -CH_2_CH = CC*H*_3_CH_3_); 1.61 (s, 3H, -CH_2_CH = CCH_3_C*H*_3_). ^13^C-NMR (DMSO-*d*_6_) δ ppm: 17.7 (-CH = C*C*H_3_CH_3_); 21.0 (-*C*H_2_CH = CCH_3_CH_3_); 25.5 (CH_2_CH = CCH_3_*C*H_3_); 93.4 (Ar*C*-8); 103.7 (Ar*C*-4a); 105.0 (Ar*C*-3); 111.2 (Ar*C*-6); 122.1 (*C*H = C(CH_3_)_2_); 126.4 (Ar*C*-2′,6′); 129.1 (Ar*C*-3′,5′); 130.7 (CH = *C*(CH_3_)_2_); 130.8 (Ar*C*-1′); 131.9 (Ar*C*-4′); 155.2 (Ar*C*-8a); 158.3 (Ar*C*-5); 162.2 (Ar*C*-7); 162.9 (Ar*C*-2); 181.9 (4-*C* = O).

5,7-dihydroxy-6-(3-methyl-2-buten-1-yl)-2-(4-hydroxyphenyl)chroman-4-one (**11**) and 5,7-dihydroxy-8-(3-methyl-2-buten-1-yl)-2-(4-hydroxyphenyl)chroman-4-one (**12**) were obtained as racemic mixtures because these were synthesized from (±) naringenin (**4**) (3.7 mmol) commercially available, 3-methyl-2-buten-1-ol (**6**) (7.3 mmol), and ZnCl_2_ (7.3 mmol) as described above. The crude mixture was purified using ethyl acetate-hexane in a gradient system (0 → 70% of ethyl acetate) as the mobile phase to afford **11** as a yellow powder (378 mg, 30%); mp: 206–207 °C (lit. [52] 212–214 °C); HRMS *m*/*z*, observed: 341.1384; C_20_H_21_O_6_ [M + H]^+^ requires: 341.1389. IR (KBr): ν_max_ cm^−1^: 3420, 2960, 2926, 2857, 1728, 1633, 1586, 1519, 1457, 1384, 1296. ^1^H-NMR (DMSO-*d*_6_) δ ppm: 12.40 (s, 1H, Ar-O*H*-5); 10.74 (br. s, 1H, ArO*H*-7); 9.55 (br. s, 1H, ArO*H*-4′) 7.29 (d, 2H, *J* = 8.4 Hz, Ar*H*-2′,6′-); 6.78 (d, J = 8.4 Hz, 2H, Ar*H*-3′,5′); 5.94 (s, 1H, Ar*H*-8); 5.39 (dd, 1H, *J_1_* = 12.5 Hz, *J_2_* = 2.6 Hz, C*H*-2); 5.11 (br. t, 1H, C*H* = C(CH_3_)_2_); 3.22 (dd, 1H, *J_1_* = 17.1Hz, *J_2_* = 12.7 Hz, CH*H*-3); 3.10 (d, 2H, *J* = 7.0 Hz, C*H*_2_CH = C(CH_3_)_2_); 2.66 (dd, 1H, *J_1_* = 14.3 Hz, *J_2_* = 2.9 Hz, C*H*H-3); 1.68 (s, 3H, CH_2_CH = CC*H*_3_CH_3_); 1.60 (s, 3H, CH_2_CH = CCH_3_C*H*_3_). ^13^C-NMR (DMSO-*d*_6_) δ ppm: 17.6 (-CH = C*C*H_3_CH_3_); 20.6 (-*C*H_2_CH = CCH_3_CH_3_); 25.4 (-CH_2_CH = CCH_3_*C*H_3_); 42.0 (*C*H_2_-3); 78.3 (Ar*C*-2); 94.3 (Ar*C*-8); 101.5 (Ar*C*-4a); 107.5 (Ar*C*-6); 115.1 (Ar*C*-3′,5′); 122.6 (*C*H = C(CH_3_)_2_); 128.3 (Ar*C*-2′,6′); 128.98 (Ar*C*-1′); 130.2 (CH = *C*(CH_3_)_2_); 157.7 (Ar*C*-4′); 160.5 (Ar*C*-7,8a); 164.3 (Ar*C*-5); 196.4 (4-*C* = O).

Compound **12** was obtained as a yellow powder (453 mg, 36%) mp: 193 °C (lit. [53] 183–184 °C, ±-8-prenylnaringenin); HRMS *m*/*z*, observed: 339.1236; C_20_H_19_O_5_ [M–H]^−^ requires: 339.1232. IR (KBr): ν_max_ cm^−1^: 3169, 2966, 2912, 1635, 1519, 1439, 1383, 1347. ^1^H-NMR (DMSO-*d*_6_) δ ppm: 12.09 (s, 1H, Ar-O*H*-5); 10.75 (s, 1H, ArO*H*-7); 9.56 (s, 1H, ArO*H*-4′) 7.30 (d, 2H, *J* = 8.4 Hz, Ar*H*-2′,6′); 6.78 (d, *J* = 8.4 Hz, 2H, Ar*H*-3′,5′); 5.95 (s, 1H, Ar*H*-6); 5.40 (dd, 1H, *J_1_* = 12.4 Hz, *J_2_* = 2.7 Hz, C*H*-2); 5.07 (br. t, 1H, C*H* = C(CH_3_)_2_); 3.19 (dd, 1H, *J_1_* = 17.1 Hz, *J_2_* = 12.6 Hz, CH*H*-3); 3.06 (d, 2H, *J* = 7.1 Hz, C*H*_2_CH = C(CH_3_)_2_); 2.70 (dd, 1H, *J_1_* = 17.2 Hz, *J_2_* = 2.7 Hz, C*H*H-3); 1.57 (s, 3H, CH_2_CH = CC*H*_3_CH_3_); 1.52 (s, 3H, CH_2_CH = CCH_3_C*H*_3_). ^13^C-NMR (DMSO-*d*_6_) δ ppm: 17.6 (-CH = C*C*H_3_CH_3_); 21.2 (-*C*H_2_CH = CCH_3_CH_3_); 25.5 (-CH_2_CH = CCH_3_*C*H_3_); 41.9 (*C*H_2_-3); 78.2 (Ar*C*-2); 95.3 (Ar*C*-6); 101.7 (Ar*C*-4a); 106.9 (Ar*C*-8); 115.1 (Ar*C*-3′,5′); 122.6 (*C*H = C(CH_3_)_2_); 128.0 (Ar*C*-2′,6′); 129.2 (Ar*C*-1′); 130.1 (CH = *C*(CH_3_)_2_); 157.5 (Ar*C*-4′); 159.7 (Ar*C*-8a); 161.1 (Ar*C*-5); 164.3 (Ar*C*-7); 196.6 (4-*C* = O).

2-(3,4-dihydroxy-5-(3′-methyl-2′-buten-1′-yl)phenyl)-3,5,7-trihydroxy-4*H*-chromen-4-one (**9**; Uralenol) was obtained from quercetin (**2**) (3.3 mmol), 3-methyl-2-buten-1-ol (**6**) (13.2 mmol), and ZnCl_2_ (13.2 mmol) as described above. The crude mixture was purified using ethyl acetate-hexane in a gradient system (0 → 80% of ethyl acetate) as the mobile phase to afford the title compound as a pale green powder (417 mg, 34%); mp: 205–206 °C (lit. [54] 176–178 °C); HRMS *m*/*z*, observed: 369.0974; C_20_H_17_O_7_ [M–H]^−^ requires: 369.0972. IR (KBr): ν_max_ cm^−1^: 3599, 3390, 2969, 2914, 1655, 1635, 1601, 1566, 1518, 1462, 1363, 1313, 1283, 1247, 1167. ^1^H-NMR (DMSO-*d*_6_) δ ppm: 12.52 (s, 1H, ArO*H*-5); 10.75 (br. s, 1H, ArO*H*-7); 9.31 (s, 1H, ArO*H*-3′); 9.03 (s, 1H, ArO*H*-4′); 8.93 (s, 1H, ArO*H*-3); 6.83 (s, 1H, Ar*H*-2′); 6.66 (s, 1H, Ar*H*-6′); 6.29 (d, 1H, *J* = 1.6 Hz, Ar*H*-8); 6.18 (d, 1H, *J* = 1.5 Hz, Ar*H*-6); 5.10 (br. t, 1H, C*H* = C(CH_3_)_2_); 3.12 (d, 2H, *J* = 7.0 Hz, C*H*_2_CH = C(CH_3_)_2_); 1.56 (s, 3H, CH_2_CH = CC*H*_3_CH_3_); 1.47 (s, 3H, CH_2_CH = CCH_3_C*H*_3_). ^13^C-NMR (DMSO-*d*_6_) δ ppm: 17.5 (CH = C*C*H_3_CH_3_); 25.4 (CH_2_CH = CCH_3_*C*H_3_); 31.1 (*C*H_2_CH = CCH_3_CH_3_); 93.3 (Ar*C*-8); 98.2 (Ar*C*-6); 103.5 (Ar*C*-4a); 116.5 (Ar*C*-6′); 117.1 (Ar*C*-2′); 120.3 (Ar*C*-5′); 123.5 (*C*H = C(CH_3_)_2_); 131.0 (CH = *C*(CH_3_)_2_); 132.2 (Ar*C*-3′); 136.3 (Ar*C*-2); 142.9 (Ar*C*-1′); 147.1 (Ar*C*-4′); 150.0 (Ar*C*-3); 156.7 (Ar*C*-8a); 160.9 (Ar*C*-5); 163.8 (Ar*C*-7); 176.2 (4-*C* = O).

2-(3,4-dihydroxy-5-(3′-methyl-2′-buten-1′-yl)phenyl)-3,7-dihydroxy-4*H*-chromen-4-one (**10**) was obtained from fisetin (**3**) (1.75 mmol), 3-methyl-2-buten-1-ol (**6**) (7 mmol), and ZnCl_2_ (7 mmol) as described above. The crude mixture was purified using ethyl acetate-hexane in a gradient system (0 → 80% of ethyl acetate) as the mobile phase to afford the title compound as a dark green powder (155 mg, 25%); mp: 229–230 °C; HRMS *m*/*z*, observed: 355.1176; C_18_H_19_O_6_ [M + H]^+^ requires: 355.1182. IR (KBr): ν_max_ cm^−1^: 3350, 2971, 2926, 2855, 1698, 1612, 1596, 1508, 1458, 1417, 1272. ^1^H-NMR (DMSO- *d*_6_) δ ppm: 10.68 (br. s, 1H, ArO*H*-7); 9.25 (s, 1H, ArO*H*-3′); 9.01 (s, 1H, ArO*H*-4′); 8.62 (s, 1H, ArO*H*-3); 7.93 (d, 1H, *J* = 8.6 Hz, Ar*H*-5); 6.89 (d, 1H, *J* = 8.2 Hz, Ar*H*-6); 6.83 (s, 1H, Ar*H*-2′); 6.78 (s, 1H, Ar*H*-8); 6.66 (s, 1H, Ar*H*-6′); 5.11 (br. t, 1H, C*H* = C(CH_3_)_2_); 3.12 (d, 2H, *J* = 6.8 Hz, C*H*_2_CH = C(CH_3_)_2_); 1.54 (s, 3H, CH_2_CH = CC*H*_3_CH_3_); 1.44 (s, 3H, CH_2_CH = CCH_3_C*H*_3_). ^13^C-NMR (DMSO- *d*_6_) δ ppm: 17.5 (CH = C*C*H_3_CH_3_); 25.4 (CH_2_CH = CCH_3_*C*H_3_); 31.2 (*C*H_2_CH = CCH_3_CH_3_); 101.8 (Ar*C*-8);114,6 (Ar*C*-6); 114.8 (Ar*C*-4a); 116.4 (Ar*C*-6′); 117.1 (Ar*C*-2′); 120.9 (Ar*C*-5′); 123.6 (*C*H = C(CH_3_)_2_); 126.5 (Ar*C*-5); 130.8 (CH = *C*(CH_3_)_2_); 132.0 (Ar*C*-3′); 137.8 (Ar*C*-1′); 142.8 (Ar*C*-2); 146.8 (Ar*C*-4′); 148.1 (Ar*C*-3); 156.7 (Ar*C*-8a); 162.1 (Ar*C*-7); 172.1 (4-*C* = O).

3-(4-hydroxyphenyl)-8-(3-methyl-2-buten-1-yl)-5,7-dihydroxy-4*H*-chromen-4-one (**13**; lupiwighteone) was obtained from genistein (**5**) (1.9 mmol), 3-methyl-2-buten-1-ol (**6**) (7.4 mmol), and ZnCl_2_ (7.4 mmol) as described above. The crude mixture was purified using ethyl acetate-hexane in a gradient system (0 → 80% of ethyl acetate) as the mobile phase to afford the title compound as a white powder (231 mg, 36 %); mp: 140 °C (lit. [55] 133–135 °C); HRMS *m*/*z*, observed: 339.1228; C_20_H_19_O_5_ [M + H]^+^ requires: 339.1232. IR (KBr): ν_max_ cm^−1^: 3350, 3174, 2921, 1705, 1651, 1613, 1570, 1513, 1428, 1364, 1301, 1257, 1198. ^1^H-NMR (DMSO-*d*_6_): δ ppm: 12.98 (s, 1H, ArO*H*-5); 9.58 (br. s, 1H, ArO*H*-4′); 8.38 (s, 1H, Ar*H*-2); 7.37 (d, 2H, *J* = 8.6 Hz, Ar*H*-2′,6′); 6.80 (d, 2H, *J* = 8.6 Hz, Ar*H*-3′,5′); 6.30 (s, 1H, Ar*H*-6); 5.14 (br. t, 1H, C*H* = C(CH_3_)_2_); 3,34 (overload with H_2_O, C*H*_2_CH = C(CH_3_)_2_); 1.74 (s, 3H, -CH_2_CH = CC*H*_3_CH_3_); 1.62 (s, 3H, -CH_2_CH = CCH_3_C*H*_3_). ^13^C-NMR (DMSO- *d*_6_) δ ppm: 17.7 (-CH = C*C*H_3_CH_3_); 21.0 (-*C*H_2_CH = CCH_3_CH_3_); 25.4 (CH_2_CH = CCH_3_*C*H_3_); 98.5 (Ar*C*-6); 104.4 (Ar*C*-4a); 105.8 (Ar*C*-8); 115.0 (Ar*C*-3′,5′); 121.3 (Ar*C*-1′); 121.9 (Ar*C*-3); 122.1 (*C*H = C(CH_3_)_2_); 130.1 (Ar*C*-2′,6′); 131.0 (CH = *C*(CH_3_)_2_); 154.0 (Ar*C*-2); 154.8 (Ar*C*-8a); 157.3 (Ar*C*-4′); 159.5 (Ar*C*-5); 161.7 (Ar*C*-7); 180.5 (4-*C* = O).

### 3.2. Biological Assays

#### 3.2.1. In Vitro Antibacterial Activity Assays: Human Pathogens

##### Minimum Inhibitory Concentration Assay

Two clinical isolates of methicillin-resistant *S. aureus* (622-4 and 97-7) and one clinical isolate of *E. coli* 33.1 were kindly donated by Dr. Marcela Wilkens from Universidad de Santiago de Chile. *S. aureus* (NCTC8325-4) and *E. coli* (ATCC25922) were used as the control strains. The antimicrobial activities of the isolated compounds against *E. coli* (ATCC25922), multi-resistant *E. coli* (33.1), *S. aureus* (NCTC8325-4), and methicillin-resistant *S. aureus* (MRSA) 97-7 strains were assessed using the Clinical and Laboratory Standards Institute (CLSI) microdilution method [56]. Briefly, stock solutions (5 mg/mL) of compounds in DMSO were two-fold diluted in Mueller–Hinton broth (MHB). The final concentration of DMSO was ≤2.5% and does not affect the microbial growth. The obtained solution was added to MHB and serially two-fold diluted in a 96-well microplate. A sample of 100 mL of inoculum 1.5 × 10^6^ colony-forming units (CFU) ml^−1^ in MHB was added and then incubated at 37 °C for 18 h. The assay was repeated three times. Wells containing MHB, 100 mL of inoculum, and DMSO served as negative controls. The MIC was defined as the lowest concentration of compounds resulting in the complete inhibition of visible growth [57].

##### Checkerboard Dilution Test

The assessments of synergy of prenylated flavonoids **11** and **12** combined with methicillin were investigated with the checkerboard method [57]. Briefly, the uppermost row (A) of a 96-well microtiter plate contained substance **X** in a concentration of about four times the expected MIC of the microorganism examined. Each following row (B–H) contained half the concentration of the previous one. The same procedure was carried out along the columns (1–12) with substance Y, but not necessarily with the same starting concentration. Therefore, each well contained a unique combination of the two substances (X and Y). Lastly, 100 μL of Mueller–Hinton broth containing about 10^5^ (CFU/mL) were added to the wells and incubated at 37 °C for 24 h. The concentrations of the first wells without visible growth along the stepwise boundary between inhibition and growth were used to calculate the FICI values. The fractional inhibitory concentration index (FICI) was calculated as follows: FICI = fractional inhibitory concentration of A (FIC_A_) + (FIC_B_) = (MIC of drug A in combination/MIC of drug A alone) + (MIC of drug B in combination/MIC of drug B alone). FIC index was interpreted as follows: synergy, <0.5; partial synergy, 0.5–0.75; additive effect, 0.76–1.0; indifference, >1.0–4.0; antagonism, >4.0 [58].

##### Time Kill Assay

Time kill curves were conducted with a final concentration of antimicrobial agent at four times the MIC [59]. Flasks containing 50 mL of MHB with the appropriate antimicrobial agent were inoculated with 50 mL of the test organism in a logarithmic growth phase adjusted to the appropriate density. Aliquots were removed, diluted, and plated at the 0, 2, 4, 6, and 24 h time points. The plates were incubated for 24 h, and the viable counts were determined. Bactericidal activity was defined as a reduction of 99.9% (≥3 log10) of the total count of CFU/mL in the original inoculum [56]. Bacteriostatic activity was defined as the maintenance of or a reduction of less than 99.9% (3 log10) of the total count of CFU/mL in the original inoculum.

##### Determination of Reactive Oxygen Species (ROS) Intrabacterial Levels

The production of ROS by *S. aureus* ATCC 6538 after treatment with the prenylated compounds was evaluated using the peroxynitrite indicator 2′,7′-dichlorodihydrofluorescein diacetate (DCFH-DA) (Sigma–Aldrich), which can detect a broad range of ROS including nitric oxide and hydrogen peroxide. The adjusted bacterial culture (0.5 McFarland exponential phase bacteria culture) was treated with a concentration of each compound of 2 µM, using ciprofloxacin as an antibiotic control, Trolox as a ROS scavenger control, 1,4-naphtoquinone as a ROS generation control, and the presence of DCFH-DA at a final concentration of 5 mM in 0.85% saline and incubated at 37 °C aerobically at 200 rpm for 24 h. Untreated bacterial culture was served as the negative control. The fluorescence emission of DCFH-DA was measured at 525 nm using a Tecan microtiter plate reader with an excitation wavelength of 485 nm. The background fluorescence of 0.85% saline and autofluorescence of the bacterial cells incubated without the probe was measured to calculate the net fluorescence emitted from the assay itself. The experiment was conducted in triplicate [60].

#### 3.2.2. In Vitro Antifungal Activity Assays against Human Pathogens

##### Microorganisms and Media

For the antifungal evaluation, standardized strains from the American Type Culture Collection (ATCC, Manassas, VA, USA), *Centro de Referencia en Micología* (CCC), Facultad de Ciencias Bioquímicas y Farmacéuticas, Suipacha 531-(2000)-Rosario, Argentina and Malbrán Institute (IM), Av. Velez Sársfield 563, Buenos Aires, Argentina were used. The voucher specimens of standardized strains are as follows: *C. albicans* ATCC 10231, *C. neoformans* ATCC 32264, *A. flavus* ATCC 9170, *A. fumigatus* ATTC 26934, A. *niger* ATCC 9029, *T. rubrum* CCC 110, *T. mentagrophytes* ATCC 9972, and *M. gypseum* CCC 115. Clinical isolates of *C. neoformans* (*n* = 5) were provided by IM. They included five strains whose voucher specimens are presented in Table 7. Clinical isolates of *C. albicans* (*n* = 5) were provided by CCC; their voucher specimens are presented in Table 8.

Strains were grown on Sabouraud-chloramphenicol agar slants for 48 h at 30 °C, maintained on slopes of Sabouraud-dextrose agar (SDA, Oxoid) and sub-cultured every 15 days to prevent pleomorphic transformations. Inocula were obtained in accordance with reported procedures [39,40] and adjusted to 1–5 × 10^3^ cells with colony-forming units (CFUs)/mL.

##### Antifungal Susceptibility Testing

The minimum inhibitory concentration (MIC) of each compound was determined by using broth microdilution techniques according to the guidelines of the CLSI for yeasts (M27-A3) [39] and for filamentous fungi (including dermatophytes) M38-A2 [40]. MIC values were determined in RPMI-1640 (Sigma–Aldrich) and buffered to pH 7.0 with MOPS. Microtiter trays were incubated at 35 °C for yeasts and *Aspergillus* spp. and at 28–30 °C for dermatophyte strains in a moist, dark chamber. MICs were visually recorded at 48 h for yeasts, and at a time according to the control fungus growth, for the rest of the fungi. For the assay, stock solutions of pure compounds were two-fold diluted with RPMI-1640 from 250 to 0.98 µg/mL (final volume = 100 µL) and a final DMSO concentration of ≤1%. A volume of 100 µL of inoculum suspension was added to each well with the exception of the sterility control, where sterile water was added to the well instead. Terbinafine (Novartis Co, Basel, Switzerland) and amphotericin B (Sigma–Aldrich) were used as positive controls. Endpoints were defined as the lowest concentration of drug resulting in the total inhibition of visual growth compared to the growth in the control wells containing no antifungal drug.

The MFC of each compound against an isolate was determined as follows: After determining the MIC, an aliquot of 5 µL was withdrawn from each clear well of the microtiter tray and plated onto a 150 mm RPMI-1640 agar plate buffered with MOPS (Remel Inc., Lenexa, KS, USA). Inoculated plates were incubated at 30 °C, and the MFC was recorded after 48 h. The MFC was defined as the lowest concentration of each compound that resulted in the total inhibition of visible growth.

#### 3.2.3. In Vitro Antibacterial Activity against Plant Pathogens

Broth microdilution methods were used to evaluate the effects of the compounds on the growth of *P. carotovorum* (NCPPB 312), *A. tumefaciens* (strain C58C1), and *P. syringae* (NCPPB 281). Bacteria were grown in sterile tubes with 10 mL of Mueller–Hinton (MH) medium and incubated at 27 °C for 12 h with shaking to produce an initial culture. The antimicrobial activity was evaluated by observing the growth response of both microorganisms in samples with different concentrations of the compounds [61,62,63]. All assays were performed on sterile 96-well microplates with a final volume of 200 µL containing Mueller–Hinton broth (MH) inoculated with 1 µL of bacterial suspension (10^5^–10^6^ UFC/mL, initial culture) in the presence of different concentrations of test compounds (3.9, 7.8, 15.6, 31.3, 62.5, 125, and 250 µM). MH was used as the negative control [C(−)], and MH with streptomycin [64] was used as the positive control [C(+)]. The plates were incubated for 7 h at 27 °C. Bacterial growth was monitored by measuring the optical density at 595 nm every hour with a microplate reader. All tests were performed in 10 repetitions for each microorganism evaluated. Bacterial growth was shown as the arithmetic mean expressed in terms of the negative control (100% growth). The lowest concentration of the compound preventing the appearance of turbidity was considered to be the minimal inhibitory concentration (MIC).

The first experiment (first kinetic assay) in which the compound was exposed to the bacterial cultures in MH was carried out over a period of 6 h. Subsequently, to determine the minimal bactericidal concentration (MBC), a second experiment (second kinetic assay) was conducted; this experiment involved taking inoculum from the first kinetic assay and adding it to MH in a new 96-well microplate containing culture medium, which was then cultured for 7 h. The aim of this second culture (second kinetic assay) was to determine whether the compounds have bactericidal or bacteriostatic properties [65].

#### 3.2.4. Statistics

The Mann–Whitney U test (*) with *p* < 0.05 was performed to identify significant differences among the treatment and control groups and between the two treatments.

### 3.3. In Vitro 5-LOX Enzyme Inhibition Assay

The commercially available enzyme by Cayman Chemicals Inc., Ann Arbor, MI, USA, was diluted (1:500) in the assay buffer (HEPES 50 mM, EDTA 2 mM, ATP 10 µM, and CaCl_2_ 10 µM at pH 7.5) and mixed with 10 µM H_2_DCFDA dye in the reaction mixture and incubated for 15 min in the assay plate. Subsequently, 280 µL of buffer was added, and 10 µL of inhibitor with a final concentration of 10 µM was placed per well in the reaction mixture and incubated for 30 min. The reaction was started by the addition of a suitable concentration of arachidonic acid (0.5 µM). The fluorescence was read in a multimode detector Synergy™ HT Multi-Mode Microplate Reader (Biotek) at 480 nm excitation/520 nm emission after 1 h of incubation at room temperature. The inhibition percentage was obtained for analysis of oxidation of H_2_DCFDA dye to the highly fluorescent 2′,7′-dichloro-fluorescein (DCF) product.

### 3.4. Docking of **11** (S and R Enantiomers) and with ABC Transporter

The prenylated flavonoids structures were built with the Molecular Operating Environment software [66]. ChelpG charges were obtained at the B3LYP/6-31G** level theory, employing the Gaussian 09 package [67]. Docking was done with the crystal structure of the ABC transporter (PDB code: 2ONJ, 3.40 Å resolution) using the AutoDock4 package [68] and a Lamarckian algorithm, assuming total flexibility of the inhibitors. The grid maps were made up first to 126 × 126 × 126 points and later to 60 × 60 × 60, with a grid-point spacing of 0.375 Å on the grid map. The AutoTors option was used to define the ligand torsions, and the docking results were then analyzed by a ranked cluster analysis, resulting in conformations with the highest overall binding energy (the most negative Gibbs free energy binding value, –ΔG).

### 3.5. General Procedure to Determine the DPPH Radical Scavenging Activity

The radical scavenging activity of the prenylated compounds and starting materials towards the 2,2-diphenyl-1-picrylhydrazyl (DPPH) radical was measured as described [18], adapted to a screen to 96-well plates. Briefly, stock solutions of each compound were prepared in methanol at a 1-mM concentration (10 mL). Dilutions (1–200 µM) were prepared from the stock solutions. Methanol (90 µL), dilutions (150 µL), and DPPH (60 µL, Sigma–Aldrich) in methanol (0.5 mM), resulting in a final concentration of 0.1 mM of DPPH were added in a 96-well plate. Methanol was used as the blank sample. The mixtures were left for 30 min at room temperature, and the absorbances were then measured at 517 nm. Trolox was used as the standard antioxidant. The radical scavenging activity was calculated as follows: % Inhibition = [(blank absorbance − sample absorbance)/blank absorbance] × 100. The mean of three IC_50_ (concentration causing 50% inhibition) values for each compound was determined graphically.

### 3.6. In Vitro Anticancer Activity of **11** and **12**

The cell lines used in this work included MDA-MB-231 human breast adenocarcinoma cells, B16-F10 mouse metastatic melanoma cells, and MEF primary mouse embryonic fibroblasts. Cells were maintained in a DMEM high glucose medium (Mediatech, Manassas, VA, USA) supplemented with 10% (MDA-MB-231 and B16-F10) or 15% (MEF) heat-inactivated fetal bovine serum (HyClone Laboratories), 100 IU/mL penicillin, and 100 μg/mL streptomycin and maintained at 37 °C in a 5% CO_2_ humidified atmosphere. Cell viability was measured using CyQuant® Direct Cell Proliferation Assay Kit (Life Technologies) following the manufacturer’s instruction. Briefly, 5.000 cells/well were seeded onto a flat-bottomed 96-well plate in a 200 μL final volume. Six hours after seeding, the culture medium was replaced with the medium containing the tested compounds at concentrations ranging from 0 to 100 μM dissolved in DMSO (a 0.1% final concentration) for 72 h. The concentrations used to calculate the IC_50_ values were 100, 30, 10, 3, 1, 0.3, 0.1, 0.01, and 0 μM. Untreated cells (medium containing 0.1% DMSO) were used as controls. At the end of the incubation, 100 μL of culture medium was removed from each experimental well and replaced by 2× detection reagent. Cells were incubated for 1 h, and fluorescence emission was measured at 535 nm with excitation at 480 nm in a microplate reader (Infinite 200 PRO, Tecan). At least four independent experiments were performed for each concentration. The results from each experiment were transformed to a percentage of controls, and the IC_50_ values were graphically obtained from the dose–response curves. The IC_50_ value was obtained adjusting the dose–response curve to sigmoidal curves (variable slope), generated using GraphPad Prisma 6.0 software [69].

## 4. Conclusions

Our results demonstrated that the 6-prenylated-(±)-naringenin (**11**), 8-prenylated-(±)-naringenin (**12**) and the 8-prenylated genistein (**13**) are active molecules against *Staphylococcus aureus*-resistant bacteria with a strong synergistically effect in combination with commercial antibiotics vancomycin, ciprofloxacin, and methicillin, enhancing their effects in a factor of 10–100 times against drug-resistant bacteria. Compound **12** is the most active against the dermatophytes *T. rubrum*, *T. mentagrophytes*, and *M. gypseum fungus.* Prenylation substitution was shown to be fundamental for these biological activities, specifically at positions 6 and 8 of the (±)-naringenin (**11** and **12**, respectively); however, others structural requirements are important as well, such as hydroxylation at positions 5, 7 of ring A, and 4′ of ring B. According to the docking results of **11** with ATP-binding cassette transporter, the *S* isomer showed better interaction energy than the *R* isomer. It is relevant to continue this study with enantiomerically pure isomers of prenylated flavanones **11** and **12**.

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
