# Peer review of "Prenylated Flavonoids with Potential Antimicrobial Activity: Synthesis, Biological Activity, and In Silico Study"

_ijms, 2021, doi:10.3390/ijms22115472_

Round 1

Reviewer 1 Report

The manuscript is well written and presented and after minor revision it can be considered for publication. The presented results could be valuable for the development of new antibacterial drugs against MRSA infection.

From the perspective of chemical synthesis, the novelty of the synthetic route chosen is not clearly presented and I suggest emphasizing this aspect.

Queries to be checked:

-Line 95, 108, 126, 134, 144: is the capital letter for the ‘position’ and ‘compound’ words necessary?

-Scheme 1: please correct the letter i above the arrow and replace it with the letter a used in the scheme description. Number 6 (prenol) must be written in bold.

- in line 158 it is described the ability of compound 13 to reduce the MIC of ciprofloxacin but Table 3 does not report the experimental value found on MRSA strains 97-7.

- in 3.1.1. paragraph, line 376, the use of CDCl3 and DMSO-d6 solvents for NMR studies is described but all 1H and 13C NMR spectra reported were recorded using only DMSO-d6.

- line 387 iupac name (3-methyl-2-buten-1-ol) for compound 6 is reported but in all main text prenol common name is used. Please try to conform the text.

- in line 433 the value of 5.94 ppm has been attributed to ArH-6 while it should be attributed to ArH-8.

- in line 445 the value of 5.96 ppm has been attributed to ArH-8 while it should be attributed to ArH-6.

Author Response

Dear Referee We have responded to your comments in the attached file. We appreciate your contributions to our work.
Best regards
Authors

Reviewer 2 Report

Osorio et al reported the synthesis of a small set of 7 novel prenylated flavinoids aiming at exploring a prenylated flavinoid analogue with potent antimicrobial activity. To achieve this aim, the authors screened the antimicrobial activity of synthesized compounds, which revealed that the prenylation of flavinoids is critical structural feature to achieve potential biological activity. Among tested compounds, compound 9, 11,12 showed potential antifungal activity against investigated human pathogenic fungi. On the other hand, compounds 11, 12, and 13 showed to be promising antibacterial activities against  Staphylococcus aureus MRSA. These results indicate that compounds 11 and 12 could be considered as potential antimicrobial hemisynthetic flavinoid. Interestingly, investigating the cytotoxicity of compounds 11 and 12 against cancer and normal cell lines showed that these compounds exhibited low cytotoxicity effect. To get deep insight the target of these compds, the authors have performed a molecular docking study which showed that compd 11 could be a good binder to the ATP-binding transporter. Overall, this is an interesting study which represent the potency of hemisynthesized flavinoids as promising antimicrobial agents. The study is well designed and performed, however, the authors need to address some concerns before this study would be suitable for publication.

1- I would suggest that the authors specify more the title of the paper as it is appear too general. it could be''Novel prenylated flavinoids with potential antimicrobial activity; synthesis, biological activity, in silico study''

2- the abstract should be modified. It is better to discuss/state the structural variation related to activity and ROS activity, than presenting MIC values. This would improve significance of presented work and motivate the reader to go through theMS. Additionally, the conclusion should be better summarize the study and highlight more the new findings.

3- the introduction part should also include/cite/summarize the previous trials for synthesized prenylated flavinoids and their activity.

4- please include the 2D-NMR which showed the specificity of prenylation of flavinoids.

5- H-NMR for compds 9, 10 showed a singlet peak at at 2ppm which is integrated to 2H or 6H, what these H assigned for?

6- What is the stereopurity of compd 4, 11, and 12? racemic? the authors should perform chiral HPLC for these compds to clearly state the purity.

7- HNMR for compd 12 showed some tiny peak for the AROH which is quite weird to understand. Can the authors explain it?

8- the purity and characterization of starting flavinoids are mentioned in the material and methods section neither the main text. Please, give some details about it.

9- the synthetic part is poorly described. please detailed insights the synthesis of each type of flavinoid derivative, source of starting flavinoids and the obtained mixture, method of separation, tics to isolate it?

10- In table 4. It is stated that the combined antibiotic with compds reduce MIC 100 fold, which id wrong it should be 10 fold.

11- The major drawback of this study is the docking study. In the presented form, it is not informative and usless. The authors need to presented in more detailed and present more data. The figures are also not well presented and does not really provide any kind of information. The authors has to check the style or format of 3D shots of MD. Its also accepted to provide the 2D shots. Detailed about the binding affinity, essential amino acids in the binding pocket, the H-bonds length, the formed H-bonding, the hydrophobic interactions...should be presented. It would be interesting that the authors show the MD of 11 in compare to e.g 13 (inactive one) or 12, this would give deep insight the difference in binding mode and the ability to bind to essential amino acids residue in the cavity and their in vitro antimicrobial activity. Additionally, in the methods part the default parameters should be mentioned.

Author Response

(The authors gave the same response as above.)

Reviewer 3 Report

Manuscript ID: ijms-1218291

          The article titled “Synthesis and Antimicrobial Activity of Prenylated Flavonoids” by Mauricio Osorio & Yesseny Vásquez-Martínez et al have demonstrated the synthesis and evaluated the antimicrobial activity of prenylated flavonoids. Combination of these flavonoids with commercial antibiotics synergistically enhanced their antibacterial activity. These prenylated flavonoids were non-cytotoxic to non-cancerous MEF cells and had low toxicity to cancerous cells.

          This manuscript is well written and is scientifically sound. A lot of chemical and biochemical data is generated and therefore this makes it valuable to the scientific community working in the field of natural product derivatives for antimicrobial drug development. Therefore, this paper does merit acceptance in IJMS.

General comments-

          In the abstract “Compounds 11, 12, and 13 (all belonging to flavanones)” to be added.

          Commonly used names of all flavonoids should be included wherever possible as indicated in example above.

          In line 38, you have mentioned only prenyl and geranyl, but even lavandulyl have shown bioactivity; see Oh I, Yang WY, Chung SC, et al. (2011). In vitro sortase A inhibitory and antimicrobial activity of flavonoids isolated from the roots of Sophora flavescens. Arch Pharm Res 34:217–22.

          Insert line 39-40 “Since prenylated flavonoids are generally more bioactive than their non-prenylated precursors, they are of interest as lead compounds for producing new drugs.” before line 43-46. This prevents from including additional reference for this statement, as refs 2-9 are self-explanatory.

Ref 8 can be updated to additional latest ref from same author lab- https://link.springer.com/article/10.1007/s11101-019-09641-z

On a general note, several prenylated flavonoids have shown better activity when connected to the 8-position (see example refs- http://dx.doi.org/10.1080/10408398.2015.1074532; https://doi.org/10.3109/13880209.2013.853809). In your case flavanones 11, 12 the prenyl group was connected at the 6-position and at the 8-position in isoflavone 13. You provide ref 27 and 28 to support your results. Additional recent literature for activities of similar substituted 6/8-prenylated flavonoids could be mentioned in the discussion.

As an example see- https://doi.org/10.1186/s12906-019-2600-y; Refer to ref no. 18 in https://doi.org/10.1038/s41598-018-27545-4

Delete lines 709-711

In the references section, include the missing DOI if any.

Author Response

(The authors gave the same response as above.)

Round 2

Reviewer 2 Report

thanks to the author for addressing all the points that have been raised. The manuscript has been significantly improved. Accordingly, I would suggest the publication of this study in the present form.